# Does Serum LH Level Influence IVF Outcomes in Women with PCOS Undergoing GnRH-Antagonist Stimulation: A Novel Indicator

**DOI:** 10.3390/jcm11164670

**Published:** 2022-08-11

**Authors:** Jing Wang, Jinli Ding, Bing Qu, Yi Zhang, Qi Zhou

**Affiliations:** 1Reproductive Medical Center, Renmin Hospital of Wuhan University & Hubei Clinic Research Center for Assisted Reproductive Technology and Embryonic Development, Wuhan 430060, China; 2Department of General Surgery, Renmin Hospital of Wuhan University, Wuhan 430060, China

**Keywords:** polycystic ovary syndrome, flexible GnRH antagonist protocol, luteinizing hormone, in vitro fertilization, intracytoplasmic sperm injection

## Abstract

Objective: To explore the influence of LH levels on the IVF/ICSI outcomes in women with PCOSundergoing GnRH-antagonist stimulation protocol. Methods: A total of 142 IVF/ICSI patients in which the females were diagnosed with PCOS and underwent GnRH-antagonist protocol for ovarian stimulation were enrolled. Patients were divided into three groups based on basal LH (bLH) level, LH level on trigger day (hLH), and the ratio of hLH/bLH. The LH levels detected on different days in the stimulation cycle as well as their relationships with the IVF/ICSI outcomes were investigated. The main outcomes we observed were the number of oocytes retrieved, the cumulative chemical pregnancy rate, clinical pregnancy rate, and live birth rate. Other factors included the number of normally fertilized oocytes (2PN), top-quality embryo rate, and total Gn dose. Results: There was no significant difference in the included outcomes and baseline characteristics among different groups based on bLH levels. When patients were grouped according to hLH levels (≤2 mIU/mL, 2–5 mIU/mL and ≥5 mIU/mL), we found decreased levels of basal FSH and LH in the group of hLH ≤ 2 mIU/mL than the other two groups. Then the ratio of hLH/bLH was calculated for each patient. Patients with hLH/bLH ≥ 1 had a higher top-quality embryo rate than those with hLH/bLH between 0.5 and 1.0. Nevertheless, the cumulative clinical pregnancy rate was significantly higher in the hLH/bLH ≤ 0.5 group than in the other two groups. Conclusions: The study proposed the hLH/bLH ratio as a potential in predicting the influence of LH level on the embryo development potential as well as pregnancy outcomes in women with PCOS undergoing GnRH-antagonist stimulation cycles.

## 1. Introduction

Polycystic Ovary Syndrome (PCOS) is a common endocrine disease that affects about 5–10% of females of reproductive age [1]. It is characterized by persistent anovulation, hyperandrogenemia, and polycystic ovarian morphology, often accompanied by insulin resistance. Insulin oversecretion promotes androgen synthesis and release in theca cells by directly improving the effectiveness of Luteinizing Hormone (LH) and indirectly increasing the pulse amplitude of LH. Based on the two cell–two gonadotropin theory, Follicle Stimulating Hormone(FSH) and LH play a role in promoting follicular growth and maturation. LH and FSH are glycoproteins synthesized and secreted by the anterior pituitary gland. In the clinic, the levels of LH and FSH are normally used as indirect evaluation indexes of ovarian reserve, whereas the ratio of LH and FSH is regarded as a predictor of ovarian responsiveness. The LH is essential for the structure and function of developing follicles. By inhibiting ovum maturation inhibitors during the early stage of normal menstruation, the LH peak may stimulate the ovum to resume meiosis and achieve final maturation by inhibiting ovum maturation inhibitors. Women with PCOS had significantly higher LH levels and significantly lower FSH levels than normal healthy women, resulting in significantly higher LH/FSH ratios, and increasing androgen synthesis that consequently leads to hyperrecruitment of oocytes [2]. Hyperandrogenemia is driven by both insulin resistance and hyperinsulinemia; it is not a necessary diagnostic condition of PCOS, but an important clinical feature of most patients [3].

Low pregnancy rates in patients with PCOS may be due to excessive LH secretion or premature LH surge before follicular maturation [4]. It was reported that LH surges occur in approximately 20 to 26 percent of women with PCOS undergoing Controlled Ovarian Stimulation (COS) cycles [5,6]. The elevated LH level in the follicular stage is associated with decreased oocyte quality, early recovery of meiosis, and premature ovulation of oocytes, resulting in a low implantation rate or increased abortion rate [7,8]. PCOS is a disease that affects women in their lives. Since women with PCOS have difficulty ovulating spontaneously, most patients need ovulation induction to achieve pregnancy. However, women with PCOS in various clinical centers, women with PCOS are at higher risk of a series of adverse pregnancy outcomes such as gestational diabetes, gestational hypertension, preeclampsia, eclampsia, premature delivery, and abortion after pregnancy than healthy people due to the uneven management level [9]. Adverse pregnancy outcomes in PCOS patients may be associated with hyperandrogenemia, hyperinsulinemia, abnormal follicular development environment, and abnormal uterine and placental factors [10].

Women with PCOS are often affected by changes in GnRH secretion pattern, abnormal negative feedback of estrogen progesterone, high androgen, hyperinsulinemia, obesity, and other factors. Serum LH levels in women with PCOS are often increased and accompanied by normal or low FSH levels. This results in an increased LH/FSH ratio, impaired follicular maturation, and even infertility [11,12,13]. Previous studies have found that early LH peaks in antagonist ovulation induction protocol influence oocyte and embryo quality and clinical pregnancy outcomes [14,15]. Nonetheless, some studies have found that an abnormal increase in LH during the ovulation process does not affect clinical outcomes; this disparity in results may be due to population bias [16]. The objective of the present study was to investigate the relationship between serum LH levels detected at different time points in COS cycles and IVF/ICSI outcomes in women with PCOS such as the number of oocytes retrieved, top-quality embryos, cumulative clinical pregnancy rate, and live birth rate.

## 2. Methods

### 2.1. Study Population

This retrospective study was approved by the Institutional Review Board of Renmin Hospital, Wuhan University. IVF/ICSI cycles were reviewed from February 2019 to November 2020 in Renmin Hospital of Wuhan University. Only women with PCOS aged between 21 and 35 who underwent GnRH-antagonist stimulation protocol were included. The diagnosis of PCOS was achieved if the patient met two of the following criteria according to the Rotterdam ESHRE/ASRM Sponsored PCOS Consensus Workshop Group [17]: (1) oligo- or anovulation, (2) clinical and/or biochemical signs of hyperandrogenism, and (3) polycystic ovarian morphology on ultrasonography. The exclusion criteria included: (1) congenital or acquired uterine anomalies; (2) history of ovarian surgery; (3) abnormal parental karyotypes or medical conditions that contraindicated assisted reproductive technology and/or pregnancy; (4) two or more previous recurrent spontaneous abortions; (5) other known endocrine disorders; (6) previous medication of combined oral contraceptive pills or glucocorticosteroids within 2–3 months before ovarian stimulation; And (7) repeated cycles, i.e., one patient with more than one time of COH.

Patients were divided according to three different strategies, namely basal serum LH level (bLH), LH level on HCG trigger day (hLH), and the ratio of hLH/bLH (Table 1).

### 2.2. Ovarian Stimulation Protocol

All patients had their first ultrasound scan on the second or third day of their menstrual cycle, and blood samples were taken on the same day and measured for serum level of FSH, LH, estradiol (E_2_), progesterone (P) and Anti-Mullerian Hormone (AMH). The flexible protocol was carried out using recombinant follicle-stimulating hormone (rFSH, Gonal-f, Merck Serono, Darmstadt, Germany) alone or in combination with hMG (Lizhu Pharmaceutical Factory, Zhuhai, China). The gonadotropin (Gn) dose was adjusted according to ovarian responses assessed by serum estrogen levels and ultrasonography. After 4–5 days of Controlled Ovarian Hyperstimulation (COH), ultrasound examination and measurement of serum FSH, LH, E_2_ and P were performed. When the leading follicle reached the size of 12–14 mm in diameter, GnRH antagonist (Cetrorelix, Merck, Kenilworth, NJ, USA) was injected at 0.25–0.5 mg/day based on the level of serum LH. When the leading follicle was 18 mm or greater, final oocyte maturation was triggered by either Human Chorionic Gonadotropin (HCG) (Lizhu Pharmaceutical Factory, China) alone or with a dual trigger comprising 2000 IU HCG and GnRH agonist (Decapeptyl 0.2 mg, Ferring International Center SA, Kiel, Germany).

### 2.3. Laboratory Procedures

The transvaginal ultrasonography was used to guide ovum pickup 34–36 h after hCG administration. Depending on the quality of sperm, oocytes were inseminated either by IVF or ICSI. Embryos were assessed and graded according to the criteria established by the Istanbul consensus workshop [18]. Routinely, two good-quality embryos were vitrified on the third day with the remaining cultured until blastocyst stage for subsequent Frozen Embryo Transfer (FET) cycles.

### 2.4. Frozen Embryo Transfer (FET) Protocol and Luteal Phase Support

We chose frozen embryo transfer cycles to calculate cumulative chemical pregnancy rate, clinical pregnancy rate, and live birth rate. In FET cycles, an artificial or natural protocol was used for endometrial preparation. Embryo transfer was performed under transabdominal sonographic guidance. For luteal support, we used intramuscular progesterone 40 mg (Xianju Pharmaceutical Factory, Taizhou, China) and oral Duphaston (Abbott Healthcare Products B.V., Weesp, The Netherlands) 30 mg once daily. If a positive pregnancy test was observed, progesterone was continuously administered until 8–10 weeks of gestation.

### 2.5. Follow-Up Protocol and Outcomes

A blood test was performed 12 days after FET. Serum β-HCG > 10 IU/L was considered as chemical pregnancy. Clinical pregnancy was defined as the presence of a gestational sac with a fetal heart under ultrasonography 30–35 days after embryo transfer. All pregnant women were monitored until live birth. The main IVF/ICSI outcomes measured include the number of oocytes retrieved, cumulative chemical pregnancy rate, clinical pregnancy rate, and live birth rate. The time period during which cumulative chemical pregnancy rate, clinical pregnancy rate, and cumulative live birth rate were calculated was until 31 July 2021. Other analyzed outcomes in the present study mainly included the number of 2PN, the ratio of top-quality embryos, and the total dosage of gonadotropins (Gn).

### 2.6. Statistical Analysis

The research results were systematized and analyzed using Statistical Package for Social Sciences (SPSS) 22.0 software. The comparison of the arithmetic means of the two samples was performed using *t*-test for independent samples or Mann–Whitney test, depending on data distribution. The comparison of the frequency of attributive features was performed by χ^2^-test. The *p* value < 0.05 was considered as statistically significant.

## 3. Results

### Patients’ Characteristics, Ovarian Stimulation Profiles and Outcomes

A total of 142 patients were included. Two (1.41%) cases developed mild to moderate OHSS symptoms after oocyte retrieval, whereas no severe OHSS case was recorded. The baseline information and outcomes were studied and compared under different grouping strategies (Table 1). Briefly, there were no statistically significant differences in terms of age and AMH levels among different groups in all of the three strategies.

There were no significant differences in baseline demographic characteristics and IVF outcomes when the basal serum LH level was taken into account (Table 2).

The levels of LH on HCG day were investigated. When compared to the other two groups, Group 1 (hLH ≤ 2 mIU/mL) had lower basal FSH and LH. The cumulative chemical pregnancy rate was higher compared with that in Group 2 (hLH = 2–5 mIU/mL) when hLH was less than 2 mIU/mL (Table 3).

Then the ratio of hLH to bLH was calculated for each individual and patients were classified based on the ratio. The lowest BMI was observed when the ratio was no more than 0.5 (Group 1). Patients in the group with a hLH/bLH ≥ 1.0 (Group 3) consumed the most dosage of Gn compared with those in the other two groups. In terms of IVF/ICSI outcomes, the top-quality embryo rate was significantly higher in Group 3 than that in Group 2, whereas the highest cumulative clinical pregnancy rate was recognized in Group 1. There was no significant difference in the cumulative live birth rate among the three groups (Table 4).

## 4. Discussion

In recent years, GnRH-antagonist therapy has been proposed as the first-line protocol for COH in women with PCOS because antagonist can clearly reduce the risk of Ovarian Hyperstimulation Syndrome (OHSS) [19,20,21]. However, excessive increase and decrease of LH are inevitable in the process of GnRH-antagonist protocol. The early LH peak during ovulation induction usually refers to the endogenous LH peak before follicular maturation or when the follicular diameter line does not reach the HCG injection standard. A key step in COH is the competitive binding of GnRH-antagonist to GnRH receptors on the pituitary gland and the inhibition of endogenous premature LH peak. Nevertheless, GnRH-antagonist can reduce but does not completely prevent the occurrence of early LH peaks, and women with PCOS with elevated basal LH levels generally have a higher incidence of early LH elevation (LH ≥ 10 U/L) during GnRH-antagonist protocol. That raises the question of how the serum LH affects the outcome of IVF/ICSI and whether there is a potential indicator to evaluate the influence.

Women with PCOS are a potential high-reaction population, an antagonist protocol being able to greatly reduce the risk of OHSS through GnRH agonist trigger, while the clinical pregnancy outcome is not affected. Therefore, at present, antagonist protocols have become the main ovulation induction protocol chosen by women with PCOS. The control of serum LH levels in the GnRH-antagonist protocol is an important factor in ovulation induction therapy, since women with PCOS are usually accompanied by higher basal LH levels and an increased probability of early LH elevation. Early LH peak can cause premature ovulation and luteinization of follicles, which can affect the number of retrieved oocytes, oocyte and embryo quality, and endometrial receptivity.

In this study, subjects were divided according to three different strategies based on LH levels detected on different days in the stimulation cycle. Generally speaking, there were statistical differences in some outcomes among different groups under each strategy. Since all the patients had frozen embryo transplantation, the effect of the hormone level on the endometrial receptivity could be ignored. The level of basal LH did not affect the number of oocytes retrieved, top-quality embryo rate, clinical pregnancy rate, and live birth rate in IVF/ICSI according to the findings. When hLH was considered, we found an increase in cumulative chemical pregnancy rate in the group of hLH no more than 2 mIU/mL. Moreover, when the bLH/hLH ratio was applied, there were more significant differences in embryo quality and pregnancy rate. Factors related to the outcomes in a FET cycle include embryo development potential, endometrium preparation protocol, demographic characteristics of patients, and underlying disease that could interfere with embryo implantation and intrauterine development. When we discussed the influence of hLH/bLH in the fresh cycle on the pregnancy outcome in the FET cycle, we assume it attributed to the embryo development potential. Even though the hLH/bLH 1.0 group had the highest rate of top-quality embryos, the pregnancy rate was highest in the <0.5 group, indicating that embryo development was highest in the <0.5 group. Embryo grading cannot be exactly the same as the embryo’s developmental potential; this may also be related to the patient’s endometrial receptivity and occult underlying diseases. The index hLH/bLH reflects the dynamic change of LH between HCG day and early follicular phase. We’ve noticed that both the <0.5 group and 0.5–1.0 group have a basal LH level at around 6 mIU/mL while that value was 4.6 mIU/mL in the ≥1.0 group, which means that the corresponding LH level on HCG day should be <3 mIU/mL, 3–6 mIU/mL, and ≥4.6 mIU/mL, separately. In that sense, the decrease in embryo development potential in the 0.5–1.0 and ≥1.0 groups may be related to the high LH exposure during stimulation. However, when analyzing the basal LH and HCG-day LH levels separately, we did not achieve such a statistically significantly difference, even though there was a trend of increase in pregnancy rate in hLH ≤ 2 group.

It was also found that when hLH/bLH was greater than 1.0, patients consumed the most gonadotropin; thus, the cost was the highest. This may be because the BMI of this group of patients was the highest, which can also indicate that patients with high BMI among women with PCOS also have relatively high LH levels and high LH level is an important cause of reproductive dysfunction in women with PCOS.

Early increase of serum LH level is usually associated with high incidence in elderly patients with low ovarian storage or patients with high response, and it should be noted that different effects of elevated serum LH level on pregnancy outcome in different groups. Although women with PCOS are also highly reactive population, their internal secretion and metabolic characteristics are different. Women with PCOS are usually have elevated basal LH levels, high androgen levels, and high insulin resistance, resulting in increased expression of insulin-like growth factor, which promotes the LH receptor expression. However, the biological activity of LH receptor and its clinical significance need to be further clarified. Currently, there is no unified standard for defining the early-onset LH peak, and most studies believe that the early-onset LH peak is defined as LH ≥ 10 U/L before follicular maturation, with or without progesterone elevation [15]. The simultaneous elevation of serum LH and progesterone levels with premature follicular luteinization usually affect oocyte and embryo quality and endometrial receptivity; however, the increased LH level usually occurs when the follicular meridian is relatively small. Because there are fewer LH receptors on follicular granulosa cells, it is impossible to respond to the increased LH level; usually, increasing the dose of antagonist can reduce LH to the normal level. Women with PCOS are predisposed to early LH elevation, particularly those with high BMI and basal LH levels. Obesity has also been linked to an early LH peak in previous studies [22].

In summary, our study showed that the fluctuation in serum LH level in GnRH-antagonist protocol could interfere with pregnancy outcomes in PCOS women receiving IVF/ICSI treatment and the ratio of hLH/bLH could be a more sensitive indicator for evaluating the LH level on the embryo development potential in women with PCOS in a GnRH-antagonist protocol. However, due to the retrospective nature of this study, particularly the small number of cases of frozen embryo transfer included, there may be selection bias, and its clinical conclusions have certain limitations. Future studies will increase the number of women with PCOS to further study the impact of LH level on outcomes of IVF/ICSI, especially cumulative live birth rate. This will better clarify its impact on the outcome of IVF/ICSI, though further studies are still needed to confirm this conclusion.

## Figures and Tables

**Table 1 jcm-11-04670-t001:** Grouping strategies.

bLH (mIU/mL)			hLH (mIU/mL)			hLH/bLH Ratio		
Group 1	Group 2	Group 3	Group 1	Group 2	Group 3	Group 1	Group 2	Group 3
≤5	5–10	≥10	≤2	2–5	≥5	≤0.5	0.5–1.0	≥1

bLH: based on basal LH, hLH: LH level on trigger day.

**Table 2 jcm-11-04670-t002:** Characteristics and outcomes of patients grouped by basal LH.

Characteristics and Outcomes	Basal LH (mIU/mL)			
	Group 1 (≤5) (n = 65)	Group 2 (5–10) (n = 54)	Group 3 (≥10) (n = 23)	*p* Value
Age	28.85 ± 3.26	29.20 ± 3.63	29.09 ± 2.84	NS
AMH (ng/mL)	7.78 ± 3.51	8.39 ± 4.03	7.81 ± 3.12	NS
BMI (Kg/m^2^)	25.19 ± 3.99	24.66 ± 3.61	23.58 ± 3.44	NS
Total Gn dose (IU)	1902.50 ± 725.96	1665.57 ± 614.60	1660.87 ± 607.53	NS
Basal FSH (mIU/mL)	6.56 ± 2.05	6.76 ± 1.58	8.38 ± 1.56	NS
Basal LH (mIU/mL)	3.32 ± 1.06	6.76 ± 1.42	12.90 ± 2.88	-
Number of oocytes retrieved	17.97 ± 8.50	16.80 ± 7.51	18.00 ± 9.12	NS
Number of 2PN (n)	10.51 ± 5.82	9.02 ± 5.00	10.96 ± 7.48	NS
Ratio of top-quality embryos (%)	65.27 ± 27.95	67.81 ± 28.80	59.92 ± 34.10	NS
Cumulative chemical pregnancy rate (%) (n)	76.92 (50/65)	83.33 (45/54)	73.91 (17/23)	NS
Cumulative clinical pregnancy rate (%) (n)	61.54 (40/65)	68.52 (37/54)	56.52 (13/23)	NS
Cumulative live birth rate (%) (n)	23.08 (15/65)	31.48 (17/54)	17.39 (4/23)	NS

NS: not significant.

**Table 3 jcm-11-04670-t003:** Characteristics and outcomes of grouped by LH on HCG day.

Characteristics and Outcomes	LH on HCG Day (mIU/mL)			
	Group 1 (≤2) (n = 68)	Group 2 (2–5) (n = 48)	Group 3 (≥5) (n = 26)	*p* Value
Age	29.03 ± 3.32	29.18 ± 3.59	28.88 ± 2.98	NS
AMH (ng/mL)	7.92 ± 3.72	7.81 ± 3.65	8.46 ± 3.61	NS
BMI (Kg/m^2^)	24.47 ± 3.96	25.21 ± 3.79	24.71 ± 3.41	NS
Total Gn dose (IU)	1872.70 ± 633.64	1650.26 ± 633.40	1898.94 ± 882.80	NS
Basal FSH (mIU/mL)	6.44 ± 1.75 ^a,b^	7.26 ± 1.92 ^a^	7.50 ± 2.07 ^b^	S
Basal LH (mIU/mL)	4.46 ± 2.41 ^c,d^	6.70 ± 3.86 ^c^	7.38 ± 4.72 ^d^	S
Number of oocytes retrieved	18.26 ± 8.61	16.31 ± 7.13	17.08 ± 8.98	NS
Number of 2PN (n)	10.63 ± 5.54	9.14 ± 5.56	9.85 ± 7.01	NS
Ratio of top-quality embryos (%) (n)	67.08 ± 25.27	61.25 ± 33.53	68.47 ± 30.02	NS
Cumulative chemical pregnancy rate (%) (n)	85.29 (58/68) ^e^	69.39 (34/49) ^e^	76.92 (20/26)	S
Cumulative clinical pregnancy rate (%) (n)	69.12 (47/68)	55.10 (27/49)	61.54 (16/26)	NS
Cumulative live birth rate (%) (n)	22.06 (15/68)	20.41 (10/49)	42.31 (11/26)	NS

^a–d^: *p* > 0.05 between the two subgroups. That is, there is no statistical difference between the two groups; ^e^: *p* < 0.05 between the two subgroups; NS: not significant; S: significant differences were recognized (*p* < 0.05).

**Table 4 jcm-11-04670-t004:** Characteristics of patients grouped by hLH/bLH.

Characteristics and Outcomes	LH on HCG Day/Basal LH (Ratio)			
	Group 1 (≤0.5) (n = 87)	Group 2 (0.5–1.0) (n = 37)	Group 3 (≥1.0) (n = 18)	*p* Value
Age	28.93 ± 3.21	29.19 ± 3.64	29.11 ± 3.38	NS
AMH (ng/mL)	8.02 ± 3.46	8.12 ± 4.05	7.79 ± 3.85	NS
BMI (Kg/m^2^)	24.23 ± 3.80 ^f^	25.11 ± 3.78	26.36 ± 3.30 ^f^	S
Total Gn dose (IU)	1707.16 ± 594.10 ^g^	1635.47 ± 559.82 ^h^	2354.72 ± 954.80 ^g,h^	S
Basal FSH (mIU/mL)	6.94 ± 1.91	6.91 ± 1.51	6.94 ± 2.64	NS
Basal LH (mIU/mL)	6.57 ± 4.07	6.02 ± 2.85	4.61 ± 3.22	NS
Number of oocytes retrieved	18.39 ± 8.67	16.78 ± 6.46	14.44 ± 8.56	NS
Number of 2PN (n)	10.64 ± 5.77	9.16 ± 5.79	8.72 ± 6.19	NS
Ratio of top-quality embryos (%)	66.68 ± 27.34	56.74 ± 34.15 ^i^	76.81 ± 23.06 ^i^	S
Cumulative chemical pregnancy rate (%) (n)	85.06 (74/87) ^j^	64.86(24/37) ^j^	77.78 (14/18)	S
Cumulative clinical pregnancy rate (%) (n)	70.11 (61/87) ^k,l^	48.65 (18/37) ^k^	55.56 (10/18) ^l^	S
Cumulative live birth rate (%) (n)	21.84 (19/87)	24.32 (9/37)	44.44 (8/18)	NS

^f–l^: *p* < 0.05 between the two subgroups; NS: not significant; S: significant differences.

## Data Availability

The datasets generated during and/or analyzed during the current study are available from the corresponding author on reasonable request.

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
