# Peer review of "Does Serum LH Level Influence IVF Outcomes in Women with PCOS Undergoing GnRH-Antagonist Stimulation: A Novel Indicator"

_jcm, 2022, doi:10.3390/jcm11164670_

Round 1

Reviewer 1 Report

  As suggested in this title and conclusion, the ratio of hLH/bLH should be considered as a potential indicator in predicting the influence of LH level on the embryo development potential and pregnancy outcomes in PCOS patients undergoing GnRH-antagonist stimulation cycles.  However, there seems to insufficient explanation or analysis on this article in details, such as 1) top -quality embryo rate was higher in hLH/bLH >1, and 2) cumulative pregnancy rate was significantly higher in hLH/bLH<0.5.

  Overall, I think it is necessary to revise this article according to LH level influence IVF outcomes in this study patients.

Author Response

Response to Review1 comments

Point 1: As suggested in this title and conclusion, the ratio of hLH/bLH should be considered as a potential indicator in predicting the influence of LH level on the embryo development potential and pregnancy outcomes in PCOS patients undergoing GnRH-antagonist stimulation cycles.  However, there seems to insufficient explanation or analysis on this article in details, such as 1) top -quality embryo rate was higher in hLH/bLH >1, and 2) cumulative pregnancy rate was significantly higher in hLH/bLH<0.5.

Overall, I think it is necessary to revise this article according to LH level influence IVF outcomes in this study patients.

Response 1:As mentioned in line 271-274, Even though the top-quality embryo rate was highest in the hLH/bLH ≥ 1.0 group, the pregnancy rate was highest in <0.5 group, indicating that embryo development was highest in < 0.5 group. Embryo grading cannot be exactly the same as the embryo's developmental potential, this may also be related to the patient's endometrial receptivity and occult underlying diseases. We have revised this part in manuscript.

Reviewer 2 Report

The authors should elaborate the rationale in suggesting LH on HCG day / Basal LH ratio. Basal LH and HCG day - LH levels can be evaluated to investigate any association with IVF/ICSI outcomes. However, the article should have a stronger foundation in recommending a ratio of these two markers for possible clinical utilization.

In discussion, the authors linked the increased amount of gonadotropin to higher overall BMI in that group, which could be linked to increased frequency of PCOS.  I believe a correlation analysis would improve the scientific base in this statement and the overall quality of the paper.

Discussion needs extensive editing in terms of proper English use.

Author Response

Response to Review2 comments

Point1: The authors should elaborate the rationale in suggesting LH on HCG day / Basal LH ratio. Basal LH and HCG day - LH levels can be evaluated to investigate any association with IVF/ICSI outcomes. However, the article should have a stronger foundation in recommending a ratio of these two markers for possible clinical utilization.

Response1: The single bLH and hLH are greatly affected by the patient's underlying conditions, the bLH in the PCOS population varies. The hLH/bLH ratio represents the change of the patient's LH during the whole process of stimulation, excluding individual differences.

Point2: In discussion, the authors linked the increased amount of gonadotropin to higher overall BMI in that group, which could be linked to increased frequency of PCOS. I believe a correlation analysis would improve the scientific base in this statement and the overall quality of the paper.

Response2: The dose of Gn is based on the age, basal antral follicles, basal FSH and body surface area of patiens. Therefore, the increased amount of gonadotropin is linked to higher overall BMI in that group. Most PCOS patients have obesity problems, so this article does not conduct a correlation analysis on this issue.

Point3: Discussion needs extensive editing in terms of proper English use.

Response3: We have polished up the full text in revised manuscript.

Reviewer 3 Report

Wang et al performed a retrospective cohort analysis on women with PCOS who underwent GnRH-antagonist stimulation with the aim of exploring the relationships between favourable outcomes with LH fluctuation.

The rationale for the study and the methods used is inadequately explained which limits the significance of the findings.  

Introduction

Line 38- Please change “polycystic ovarian changes” to “polycystic ovarian morphology”

PCOS patients to women with PCOS

Make sure to define abbreviations on the first appearance

Methods

Please explain how you came up with three categories under each group of 1-3? (i.e. why did you choose the cut-offs of ≤5 5-10 ≥10 for bLH; ≤2 2-5 ≥5 for hLH and ≤0.5 0.5-1.0 ≥1 for hLH/ bLH?

Why did you choose performing descriptive analyses on the data rather than performing regression enabling adjustment for important factors such as age and BMI?

Results

What were overall mean age and BMI of participants?

Please add the analyses used for the variables at footnotes of the tables.

Instead of using “+” for significant differences, you may want to consider “s” as opposed to NS.

Discussion

Normally, a discussion starts with information which usually goes to introduction. I expected the discussion be started with the main findings in the first paragraph followed by discussing each of the main findings along with reports from prior literature. Instead, 5 paragraphs are dedicated to give information from prior literature without clearly linking it to the findings. For a reader, like me who works in the area of PCOS but not in assisted reproductive technology, the significance of the findings would be unclear.

Author Response

Response to Review3 comments

Point1:

Introduction

Line 38- Please change “polycystic ovarian changes” to “polycystic ovarian morphology”

PCOS patients to women with PCOS

Make sure to define abbreviations on the first appearance

Response1:

  1. We have changed “polycystic ovarian changes” to “polycystic ovarian morphology” in line 38.
  2. We have changed “PCOS patients” to “women with PCOS” through full text of all as in revised manuscript.
  3. We have checked the full text of the abbreviation, make sure to have defined abbreviations on the first appearance

Point2:

Methods

Please explain how you came up with three categories under each group of 1-3? (i.e. why did you choose the cut-offs of ≤5 5-10 ≥10 for bLH; ≤2 2-5 ≥5 for hLH and ≤0.5 0.5-1.0 ≥1 for hLH/ bLH?

Why did you choose performing descriptive analyses on the data rather than performing regression enabling adjustment for important factors such as age and BMI?

Response2:

  1. We consulted the reference “Sun L, Ye J, Wang Y, Chen Q, Cai R, Fu Y, Tian H, Lyu Q, Lu X, Kuang Y. Elevated basal luteinizing hormone does not impair the outcome of human menopausal gonadotropin and medroxyprogesterone acetate treatment cycles. Sci Rep. 2018 Sep 14;8(1):13835. doi: 10.1038/s41598-018-32128-4.”. In addition, we also tried a number of different grouping strategies, when the cut-offs of ≤5 5-10 ≥10 for bLH; ≤2 2-5 ≥5 for hLH and ≤0.5 0.5-1.0 ≥1 for hLH/ bLH, we can observed some outcomes were statistically different.
  2. Because there was no significant difference in age and BMI of patients after grouping.

Point3:

Results

What were overall mean age and BMI of participants?

Please add the analyses used for the variables at footnotes of the tables.

Instead of using “+” for significant differences, you may want to consider “s” as opposed to NS.

Response3:

  1. We describe the mean age and BMI of each group, by calculating, the overall mean age were 29.08±3.02, and BMI of participants were 7.86±3.31.
  2. we have add the analyses used for the variables at footnotes of the tables.
  3. We have changed “s” instead of “+” in revised manuscript

Reviewer 4 Report

1.     The “abstract” section should be modified to make a clear and precise conclusion.

2.     Please cite the references in the first paragraph in the introduction section.

3.     Not all patients with PCOS have high LH levels compared with normal healthy women. If bLH and hLH levels were considered important to the ART result and pregnancy results, how to explain the effect of low LH for some PCOS patients?

4.     Line 55-56, 69-71: please cite the reference.

5.     In this study, patients received COH with antagonist protocol. However, some patients received rFSH only, and some received hMG, which contained the FSH AND LH. The different gonadotropin may affect the serum LH level. Besides, since all the patients received antagonist protocol, the serum LH level was sensitive to the injection of GnRH antagonist. Did the patients use Cetrorelix on the trigger day? If so, how long was the interval between the last Cetrorelix injection and the test of serum LH level?

6.     In the result section, the authors claimed that they collected 142 cycles in this study but did not tell us how many patients they recruited. It would affect the result pregnancy rate, including the cumulative chemical PR, cumulated clinical PR, and cumulative LBR analyzed in their research. The denominator of pregnancy rate should be the number of patients, not cycles. And, it is confusing about the “142 cycles”. Did it mean the cycle number of COH? The authors should re-analyze the data, especially the cumulative pregnancy rate.

7.     How many embryos were transferred during the FET cycle in each patient? D3 or D5 embryos?

8.     According to the study design and data analysis in this manuscript, it could not result in the conclusion about using serum LH level as an indicator of IVF outcome.

Author Response

Response to Review4 comments

Point1: The “abstract” section should be modified to make a clear and precise conclusion.

Response1: we have revised “abstract” section based on comments.

Point2:  Please cite the references in the first paragraph in the introduction section.

Response2: we have cited the references in line 50 and 55.

Point3: Not all patients with PCOS have high LH levels compared with normal healthy women. If bLH and hLH levels were considered important to the ART result and pregnancy results, how to explain the effect of low LH for some PCOS patients?

Response3: Indeed, not all PCOS patients will be combined with high LH level, so the hLH/bLH ratio group was set up in this study to better display the influence of dynamic changes of LH on the final outcomes of ART in the process of hyperstimulation, rather than the LH level at a certain point in time. How will the final outcome of ART be in PCOS patients with low LH level? That can be incorporated into the next part of our study.

Point4: Line 55-56, 69-71: please cite the reference.

Response4: We have cited the references in revised manuscript.

Point5:  In this study, patients received COH with antagonist protocol. However, some patients received rFSH only, and some received hMG, which contained the FSH AND LH. The different gonadotropin may affect the serum LH level. Besides, since all the patients received antagonist protocol, the serum LH level was sensitive to the injection of GnRH antagonist. Did the patients use Cetrorelix on the trigger day? If so, how long was the interval between the last Cetrorelix injection and the test of serum LH level?

Response5: In 142 patients included,they all received rFSH plus hMG, the effect on serum LH should be consistent. All patients included used Cetrorelix on the trigger day, we test serum LH level in trigger day before last Cetrorelix injection and after latest Cetrorelix injection about 21 hours.

Point6: In the result section, the authors claimed that they collected 142 cycles in this study but did not tell us how many patients they recruited. It would affect the result pregnancy rate, including the cumulative chemical PR, cumulated clinical PR, and cumulative LBR analyzed in their research. The denominator of pregnancy rate should be the number of patients, not cycles. And, it is confusing about the “142 cycles”. Did it mean the cycle number of COH? The authors should re-analyze the data, especially the cumulative pregnancy rate.

Response6: In line 155, we revised to “A total of 142 patients were included.” and add another exclusion criteria in line 99: (7) repeated cycles: one patient in second or more times of COH.

Point7: How many embryos were transferred during the FET cycle in each patient? D3 or D5 embryos?

Response7: 2 embryos were transferred transferred in each patient included, based on quality control data in our center, we all transferred D3 embryos.

Point8: According to the study design and data analysis in this manuscript, it could not result in the conclusion about using serum LH level as an indicator of IVF outcome.

Response8: It is true that the LH level at a certain time point cannot predict the outcome of IVF. In this paper, based on the dynamic LH changes, namely the ratio of bLH to hLH, the relationship between the ratio and IVF related outcome indicators was explored, and it was speculated that the ratio might affect the developmental potential of embryos. However, due to the insufficient sample size, and pregnancy outcome may be related to endometrial receptivity and patients' own latent underlying diseases in addition to embryo development potential, more influencing factors should be included in future studies for analysis.

Round 2

Reviewer 3 Report

Please update the style of your references; it was confusing with a mixture of numbers and texts!

Introduction, Page 2, line 69: change “deseased” to “decreased”, if this was what you meant.

Introduction, Page 2, lines 73-77: In fact, it is unclear what exactly increase the risk of adverse pregnancy outcomes in PCOS. Uneven management could be one of the reasons but will not fully explain the increased risk. You have somehow mentioned some of the risk factors in the following sentence. Please consider revising the sentence.

Footnotes for tables are not what those should be such as defining abbreviations and symbols, statistical methods used and significant level accepted for the analysis.

The discussion looks more like an introduction to me; at least the first five paragraphs. Authors are more trying to open a question why the study was required to be performed rather than highlighting the findings and their significance.

Author Response

Response to Review3 comments (2)

Point 1: Please update the style of your references; it was confusing with a mixture of numbers and texts!

Response 1:We have revised the format of the references in the manuscript.

Point 2: Introduction, Page 2, line 69: change “deseased” to “decreased”, if this was what you meant.

Response 2:Thank you very much for the reminder, which we have corrected in the manuscript.

Point 3: Introduction, Page 2, lines 73-77: In fact, it is unclear what exactly increase the risk of adverse pregnancy outcomes in PCOS. Uneven management could be one of the reasons but will not fully explain the increased risk. You have somehow mentioned some of the risk factors in the following sentence. Please consider revising the sentence.

Response 3:Thanks for the your comments, we have made revisions in the manuscript and provided relevant references.

Point 4: Footnotes for tables are not what those should be such as defining abbreviations and symbols, statistical methods used and significant level accepted for the analysis.

Response 4:We have updated the footnotes to the table.

Point 5: The discussion looks more like an introduction to me; at least the first five paragraphs. Authors are more trying to open a question why the study was required to be performed rather than highlighting the findings and their significance.

Response 5:We have removed the too redundant background introduction in the discussion and added some descriptions of the purpose of this study.

This manuscript is a resubmission of an earlier submission. The following is a list of the peer review reports and author responses from that submission.